# Parameters Synthesis of Na-Magadiite Materials for Water Treatment and Removal of Basic Blue-41: Properties and Single-Batch Design Adsorber

Abdulaziz M. Alanazi [1,*], Hmoud Al Dmour [2], Saheed A. Popoola [1], Hicham Oudghiri Hassani [3], Souad Rakass [4], Rawan Al-Faze [5] and Fethi Kooli [1,*]

1  Department of Chemistry, Faculty of Science, Islamic University of Madinah, Madinah 42351, Saudi Arabia; abiodun@iu.edu.sa
2  Department of Physics, Faculty of Science, Mu'tah University, Mu'tah 6170, Jordan; hmoud79@mutah.edu.jo
3  Engineering Laboratory of Organometallic, Molecular Materials and Environment (LIMOME), Faculty of Sciences, Chemistry Department, Sidi Mohamed Ben Abdellah University, P.O. Box 1796 (Atlas), Fez 30000, Morocco; oudghiri_hassani_hicham@yahoo.com
4  Laboratory of Applied Organic Chemistry (LCOA), Chemistry Department, Faculty of Sciences and Techniques, Sidi Mohamed Ben Abdellah University, Imouzzer Road, P.O. Box 2202, Fez 30000, Morocco; souad.rakass@usmba.ac.ma
5  Department of Chemistry, Faculty of Science, Taibah University, Al-Madinah Al-Munawarah 30002, Saudi Arabia
*  Correspondence: a-aziz@iu.edu.sa (A.M.A.); fethi_kooli@yahoo.com (F.K.)

**Abstract:** Na-magadiite materials were prepared from a gel containing a silica source, sodium hydroxide, and water via hydrothermal treatment at different temperatures (130 °C to 170 °C) and periods of time (1 day to 10 days). In this study, four silica sources were selected (fumed silica, colloidal silica, Ludox HS-40%, and Ludox AS-40%). Variable conditions such as sodium hydroxide and water contents were explored at a specific temperature and reaction time. The obtained materials were characterized by using X-ray diffraction (XRD), thermogravimetry differential thermal analysis TG-DTA, scanning electron microscopy with energy dispersive X-ray spectroscopy (SEM-EDX), Fourier Transform Infrared spectroscopy (FTIR), solid $^{29}$Si magic angle spinning magnetic nuclear resonance (MAS MNR, and nitrogen adsorption isotherms. A pure Na-magadiite phase was obtained from the four silica sources at a synthesis temperature of 150 °C after a period of one to two days with a characteristic basal spacing of 1.54 nm. At a longer reaction time of 3 days and a higher temperature of 170 °C, Na-kenyaite with a basal spacing of 2.01 nm was achieved, in addition to a quartz phase. The content of water or sodium hydroxide in the gel affected the nature of the prepared phases. A cauliflower-like morphology was obtained from colloidal silica sources, while a different morphology was achieved using solid fumed silica. The $^{29}$Si solid NMR confirmed the presence of $Q^3$ and $Q^4$ silicon sites in the Na-magadiite materials. The optimal Na-magadiite materials at 150 °C for 2 days were assessed for their ability to remove Basic Blue-41 dye from artificially contaminated aqueous solution. The Langmuir equation was used to estimate the maximum removal capacity. A maximum removal capacity of 219 mg/g was achieved using Na-magadiite prepared from a Ludox-HS40% silica source, and a maximum removal capacity of 167 mg/g was observed for Na-magadiite prepared from fumed silica. Basic Blue-4's removal percentage was enhanced at basic pH levels (8 to 10) to a maximum of 95%. These materials could be regenerated for seven cycles of reuse with a reduction of 27 to 40% of the original values. Therefore, Na-magadiite materials are promising and efficient removal agents for the removal of Basic Blue-41 from effluents.

**Keywords:** layered silicates; Na-magadiite; Na-kenyaite; silica source; Basic Blue-41; removal; regeneration; single-batch design; water treatment

## 1. Introduction

Jing et al. [1] portrayed magadiite as the most pervasive material on earth; it is half abandoned and half disregarded, and it is incredibly significant for the advancement to valuable materials. By searching for new applications, they worked out that the longing 'waste can be transformed into treasure'.

The polysilicate family is still gaining a lot of interest currently. One of the layered hydrous silicates is named magadiite. This material has a low cost and is considered an environmentally friendly material that is easily prepared on a laboratory scale generally under hydrothermal conditions [2,3]. In a recent review by Dos Santos et al. (2023), the authors explored, in detail, the most recent developments of magadiite for the production of multifunctional lamellar materials. Owing to the interesting and attractive properties originated from the ease exchange of the hydrated sodium cations, and to the reactivity of the interlayer silanol groups, the modification of the interlayer space with different molecules was explored in different aims [4]. Compared to natural clay minerals, magadiite exhibits a negatively layered silicate structure, with a reasonably high cation exchange capacity and a layer charge density ranging from 200 to 250 meq/100 g [5,6].

The surface modification of magadiite was explored by different researchers to make it suitable for specific applications, such as in analogy and in natural aluminosilicates [7–10]. For water treatment via an adsorption process, the cation exchange reaction was explored to prepare organomagadiites and to generate an adsorbent for negatively charged dyes (such acidic dyes) [10–12]. In the case of basic dyes, the magadiite materials were used without modification [13].

The synthesis conditions of Na-magadiites have been reported and summarized by Dos Santos et al. [4]. Synthetic magadiites are most commonly prepared via hydrothermal processes. Amorphous reagents (silica, etc.) and a mineralizing agent (usually alkyl metal hydroxide (NaOH)) are dissolved in aqueous solutions at a high pH. The mixture is mixed and conditioned in an autoclave with an inner PTFE vessel closed for the crystallization phase. The mixture is then treated for a period of time at a constant temperature. The standard method for the preparation of magadiites is the hydrothermal synthesis process. The advantages of this method include a low cost, solvent use (water), high reproducibility, and promising yields. However, certain reaction parameters (nature and concentration of the precursor substances, solvent, temperature, reaction time, etc.) play significant roles in the formation of magadiite. The influence of temperature on Na-magadiite synthesis is more important than the reaction time or reagent concentrations.

On the other hand, different silica sources were used to prepare Na-magadiite, and there was not a systematic reason as to why one source was favored over the others. For example, by using silica gel as a source of silica, a pure magadiite phase was obtained at 150 °C for 3 days [14]; however, the same phase required 8 days of reaction at 150 °C using colloidal silica (40% in water) [15]. The reaction time was reduced to 3 days using Ludox-HS40 [16]. Using an amorphous silica, the magadiite phase was obtained at 170 °C for 1 day of reaction [17]. There are limited numbers of studies related to the effect of silica sources on magadiite synthesis [18–20]. As reported for the synthesis of zeolite materials, the silica source was one the key factors that determined the properties and the morphology of the obtained products, and it can impact the synthesis time [21].

In this work, the effects of silica sources on the synthesis of Na-magadiite were investigated in a systematic manner. In this regard, four silica sources were selected. Na-magadiite was synthesized using sodium hydroxide (NaOH) and water. The hydrothermal treatment method was chosen at various temperatures and time periods. The physical and chemical properties of the as-synthesized samples were analyzed via X-ray diffraction (XRD), thermogravimetric and differential thermal analysis (TGA/DTA), X-ray fluorescence spectrometer (XRF), scanning electron microscopy (SEM) associated with an EDX system, nitrogen adsorption, Fourier Transform Infrared spectroscopy (FTIR), and solid $^{29}$Silicon MAS NMR. The application of the Na-magadiite materials was performed as a model in the water treatment to remove Basic Blue-41 (BB-41) from artificially polluted water. BB-41 is an azo dye that is

typically used for dyeing wool and silk. Among the different classes of dyes, the azo-dye class was selected because it is the most used group of synthetic dyes [22]. Furthermore, cationic dyes have strong colors, which are noticeable at very low concentrations [23]. However, due to their poor degrees of fixation on the fiber, significant quantities of color are released into the effluent. BB-41 blocks light from reaching the water, which lowers the rates of photosynthesis in phytoplankton-colored algae and has an impact on the aquatic biota. Additionally, it can have a variety of negative impacts on human health [24]. To optimize the magadiite capacity and dye removal efficiency, a set of factors were selected including the initial dye concentration, the pH value of the BB-41 solution, the mass, and the morphology of the used magadiite. These variables were investigated as separate entities. The Langmuir model was applied to assess the maximum removal capacities. The linear and non-linear isotherm analyses were used to fit the data of the BB-41 removal. The regeneration of selected spent magadiites was undertaken for seven cycles, using oxone and cobalt nitrate solution [25] and a single-batch adsorber design was proposed.

## 2. Results and Discussion

Mainly two kinds of layered silicates (Na-magadiite and Na-kenyaite) were formed during the experimental procedures. The structures of these two phases and their identifications are described. The structure of magadiite has been a subject of debate, and previous attempts to solve its structure were reported. Recently, Bernd et al. (2023) solved the mystery of the magadiite structure using 3D electron diffraction (ED) and confirmed it through Rietveld refinement against powder X-ray diffraction (PXRD) data, supported by DFT calculations [26]. The authors concluded that the structure of magadiite consists of dense silicate layers without any microchannels and with a Q4-to-Q3 ratio of 10:4. The interlayer region of the structure of magadiite resembles channel-like voids occupied by bands of interconnected $[Na(H_2O)_6]$ octahedra and is nearly identical with the interlayer region of octosilicate (RUB-18, ilerite). The PXRD pattern exhibited a strong reflection at 1.54 nm with multiple reflections. A typical PXRD pattern is presented in Figure 1a.

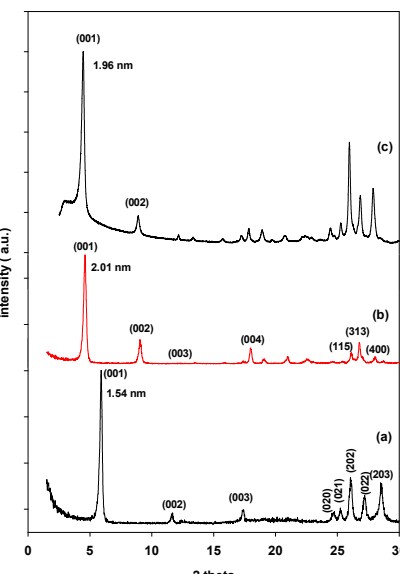

**Figure 1.** PXRD patterns of typical layered silicates, (**a**) Na-magadiite and (**b**) Na-kenyaite, prepared in this study. (**c**) corresponds to PXRD pattern of Na-kenyaite provided by Prof. Marler et al. [27].

For the kenyaite structure, recently, Marler et al. (2021) determined the structure of kenyaite using the Rietveld refinement method [27]. It consisted of stacked thick silicate layers with a thickness of 15.9 Å, with interlayer spaces occupied by water molecules and sodium ions. The dense layers possess the same topology as the layers of RUB-6, exhibit no porosity, and have a silicon $Q^4$-to-$Q^3$ ratio of 4:1. The PXRD pattern exhibited strong reflection at a distance close to 2 nm, with additional broad reflections with different half

widths. A typical PXRD pattern is presented in Figure 1b. For comparison, a reference PXRD pattern of Na-kenyaite is presented (Figure 1c).

Tables 1 and 2 summarize the details of the synthesis conditions and the resulting products obtained during the study. The variables are presented in bold characters.

**Table 1.** Crystallization of the products at various compositions of reactants and conditions of synthesis using fumed silica and Ludox-HS40.

| Run No. | Synthesis Conditions (Initial Gels) | | | | | | Products |
| --- | --- | --- | --- | --- | --- | --- | --- |
| | Silica Source | SiO$_2$ (g) | NaOH (g) | H$_2$O (g) | Temp. | Time | |
| 1 | | 45 | 4.8 | 105 | 150 | **1 day** | Mag |
| 2 | | 45 | 4.8 | 105 | 150 | **2 days** | mag (MAG-FS) |
| 3 | | 45 | 4.8 | 105 | 150 | **3 days** | Mag + ken |
| 4 | | 45 | 4.8 | 105 | 150 | **5 days** | Mag + ken |
| 5 | | 45 | 4.8 | 105 | 150 | **7 days** | Ken |
| 6 | | 45 | 4.8 | 105 | 150 | **10 days** | Ken + silica |
| 7 | | 45 | 4.8 | 105 | **130** | 2 days | Mag + amorph. silica |
| 8 | | 45 | 4.8 | 105 | **170** | 2 days | ken |
| 9 | | 45 | 4.8 | 105 | **185** | 2 days | Crystalline silica |
| 10 | **Fumed silica** | 45 | 4.8 | 105 | **190** | 2 days | Crystalline quartz |
| 11 | | 45 | 4.8 | 105 | **200** | 2 days | crystalline quartz |
| 12 | | 45 | **1.05** | 105 | 150 | 2 days | Amorph. silica |
| 13 | | 45 | **2.5** | 105 | 150 | 2 days | Mag |
| 14 | | 45 | **7.2** | 105 | 150 | 2 days | no solid was obtained |
| 15 | | 45 | **9.6** | 105 | 150 | 2 days | no solid was obtained |
| 16 | | 45 | 4.8 | **20** | 150 | 2 days | Ken |
| 17 | | 45 | 4.8 | **40** | 150 | 2 days | Ken + Mag |
| 18 | | 45 | 4.8 | **50** | 150 | 2 days | Mag |
| 19 | | 45 | 4.8 | **60** | 150 | 2 days | Mag |
| 20 | | 45 | 4.8 | 105 | 150 | **1 day** | Mag |
| 21 | | 45 | 4.8 | 105 | 150 | **2 days** | mag (MAG-HS) |
| 22 | **Ludox HS-40%** | 45 | 4.8 | 105 | 150 | **3 days** | Mag + silica |
| 23 | | 45 | 4.8 | 105 | 150 | **5 days** | Mag + ken + silica |
| 24 | | 45 | 4.8 | 105 | 150 | **7 days** | Ken + silica |

Mag: Na-magadiite, Ken: Na-Kenyaite, Tr: traces.

**Table 2.** Crystallization of the products at various compositions of reactants and conditions of synthesis using Ludox-As40 and colloidal silica.

| Run No. | Synthesis Conditions (Initial Gels) | | | | | | Products |
| --- | --- | --- | --- | --- | --- | --- | --- |
| | Silica Source | SiO$_2$ (g) | NaOH (g) | H$_2$O (g) | Temp. | Time | |
| 24 | | 45 | 4.8 | 105 | 150 | **1 day** | Mag |
| 25 | | 45 | 4.8 | 105 | 150 | **2 days** | mag (MAG-AS) |
| 26 | **Ludox AS-40%** | 45 | 4.8 | 105 | 150 | **3 days** | Mag |
| 27 | | 45 | 4.8 | 105 | 150 | **5 days** | Mag + ken + silica |
| 28 | | 45 | 4.8 | 105 | 130 | **10 days** | Ken + silica |
| 29 | | 45 | 4.8 | 105 | 150 | **1 day** | Mag |
| 30 | | 45 | 4.8 | 105 | 150 | **2 days** | mag (MAG-CS) |
| 31 | | 45 | 4.8 | 105 | 150 | **3 days** | mag + ken + tr (SiO$_2$) |
| 32 | | 45 | 4.8 | 105 | 150 | **5 days** | mag + ken + SiO$_2$ |
| 33 | | 45 | 4.8 | 105 | 150 | **7 days** | mag + ken + SiO$_2$ |
| 34 | | 45 | 4.8 | 105 | 150 | **10 days** | Ken + SiO$_2$ (quartz) |
| 35 | | 45 | 4.8 | 105 | **130** | 2 days | mag + amorp SiO$_2$ |
| 36 | **Colloidal Silica** | 45 | 4.8 | 105 | **140** | 2 days | Mag |
| 37 | | 45 | 4.8 | 105 | **170** | 2 days | Ken |
| 38 | | 45 | 4.8 | 105 | **200** | 2 days | quartz + tr (ken) |
| 39 | | 45 | **1.2** | 105 | 150 | 2 days | Amorphous silica |
| 40 | | 45 | **2.4** | 105 | 150 | 2 days | Mag |
| 41 | | 45 | **7.2** | 105 | 150 | 2 days | Mag |
| 42 | | 45 | **9.2** | 105 | 150 | 2 days | Mag |
| 43 | | 45 | 4.8 | **0** | 150 | 2 days | Mag |

Mag: Na-magadiite, Ken: Na-Kenyaite, Tr: traces; amorph: amorphous. The variables are presented in bold characters.

Qualitatively, a pure Na-magadiite phase was formed for one to two days regardless of the use of silica sources (runs 1, 2, 20, 21, 24, 25, 29, and 30). Based on the intensity of the first (001) reflection at 1.56 nm, the MAG-AS had a better degree of crystallinity. For a period of three days, only a Na-magadiite phase was obtained when Ludox-AS 40 (run 26) was used, in addition to traces of Na-kenyaite and quartz, which were obtained for longer periods of five and ten days (runs 27 and 28; Figure 2 (left)). However, different results were obtained when fumed silica was used. Mixtures of Na-magadiite and Na-kenyaite phases were obtained only for three days (run 3), as reported in Figure 2 (right). The improvement in the Na-kenyaite phase continued for a duration longer than five days (run 5). An additional quartz phase, which was related to the used silica source, was observed after ten days of reaction (run 6; Figure 2 (right)). In the case of Ludox-HS40 (runs 22–24), the PXRD data indicated that a mixture of a Na-magadiite phase and Na-kenyaite in addition to silica was obtained for five days of reaction (run 23), and only Na-kenyaite and quartz phases were detected at periods of hydrothermal treatment of ten days (run 24; Figure S2).

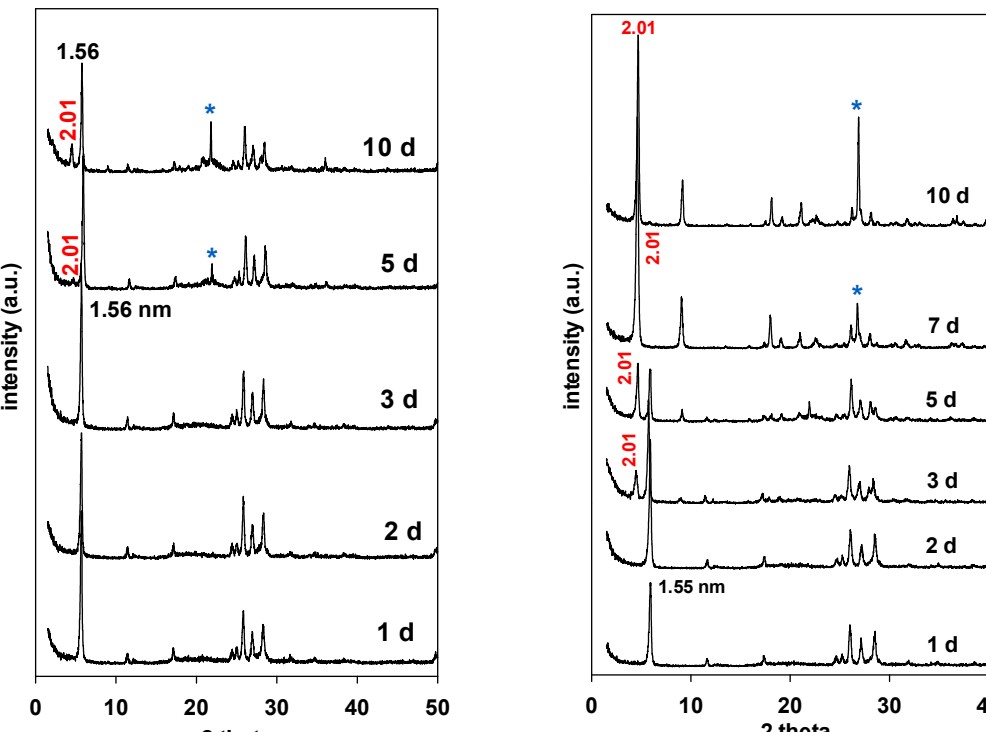

**Figure 2.** PXRD patterns of samples prepared from Ludox AS40 (**left**) and fumed silica (**right**) for different periods of time at 150 °C. (*) corresponds to silica phases.

As indicated by the PXRD patterns, the Na-magadiite and Na-kenyaite exhibited an intense reflection peak at distances ($d_{001}$) of 1.54 to 1.56 nm and 1.91 nm to 2.01 nm, respectively. This variation could be related to the method and the temperatures of drying and the temperatures that affected the content of water molecules between the interlayer spacing of the silicate layers [28]. The variation of the position of this reflection was also reported to be related to the type of exchangeable cations between the magadiite or the kenyaite layers [29,30].

## 2.1. Effect of Temperature

Two silica sources (colloidal silica and fumed silica) were investigated, as mentioned in Table 1. Figure 3 (left and right) depicts the PXRD patterns of the resulting products. Both silica sources produced a pure Na-magadiite phase in the temperature range of 130 to 150 °C (runs 7, 12, 35, and 36), while at 170 °C, only Na-kenyaite was obtained from both silica sources (runs 8 and 37), and no silicate layered materials were produced at

temperatures higher than 185 °C; a mainly quartz phase was formed for both silica sources (runs 9–11 and 38).

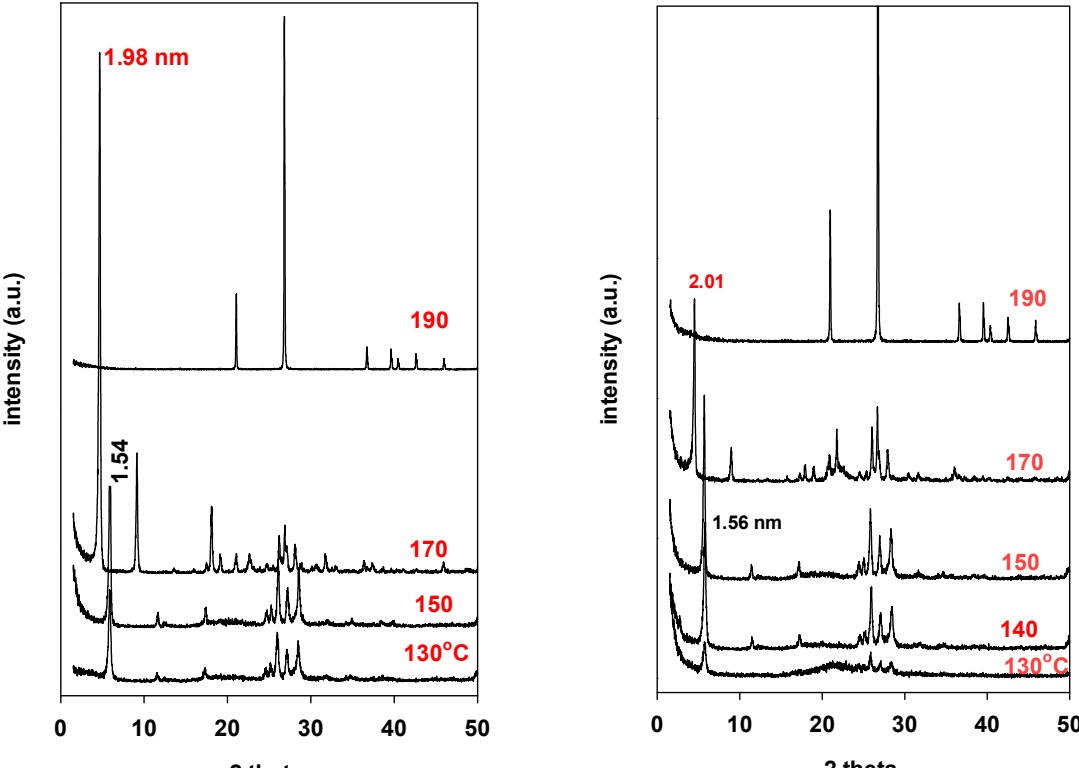

**Figure 3.** PXRD patterns of samples prepared from fumed silica (**left**) and colloidal silica (**right**) at different temperatures for a period of two days.

These data clearly indicated that the formation of Na-magadiite and Na-kenyaite depended not only on the temperature but also on the silica source. Indeed, the Na-kenyaite was obtained at 180 °C for 71 h using another silica source and gel composition [30].

## 2.2. Effect of NaOH

In this case, only fumed and colloidal silica were used as silica supplies to avoid any interference with the Na contents in the cases of Ludox-AS40 and Ludox-HS40. The temperature was kept at 150 °C for two days of hydrothermal treatment. The mass of added silica and the volumes of water were unchanged, and different masses of NaOH were dissolved in 105 g of water before adding the silica source.

When 1.2 g of NaOH was added, an amorphous silica phase was obtained independently of the used silica sources (runs 12 and 39). In the case of fumed silica, Na-magadiite was formed when 2.4 g of NaOH was added (run 13). The crystallinity of this phase was improved by increasing the amount of NaOH at 4.8 g (run 2). However, when the amount of NaOH exceeded 5 g (runs 14 and 15), all silica species were dissolved, resulting in a clear solution, and no solid was obtained (Figure 4, left).

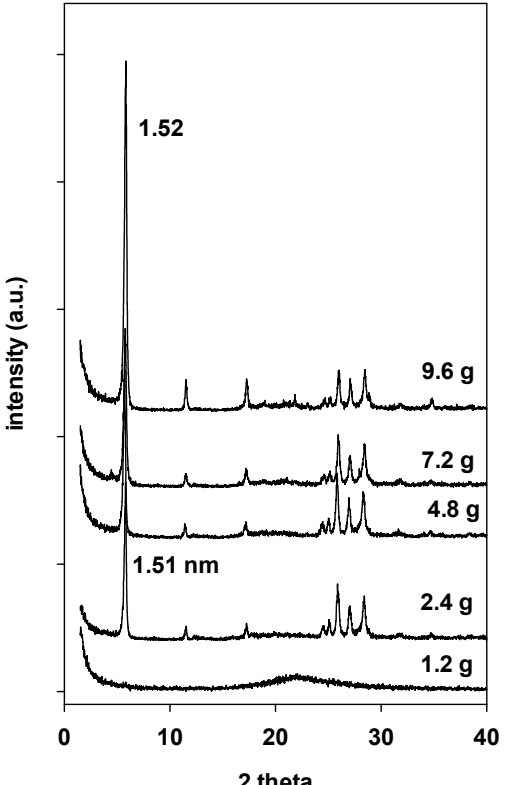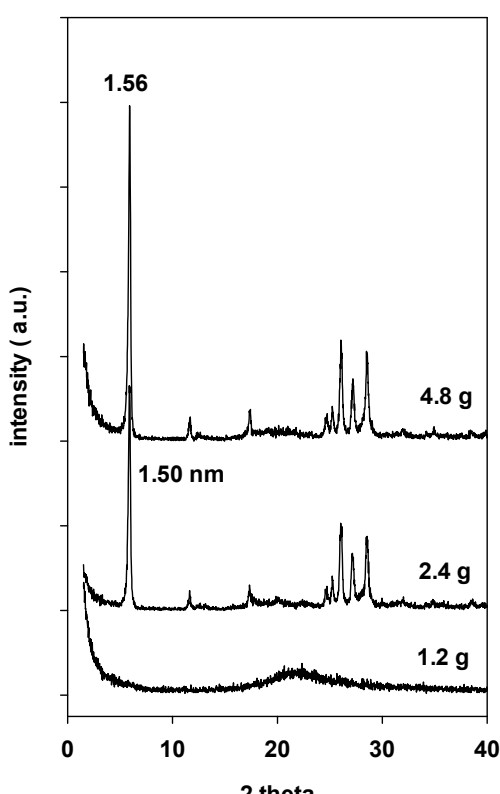

**Figure 4.** PXRD patterns of Na-magadiite prepared from colloidal silica (**left**) and fumed silica (**right**) using different amounts of NaOH.

Different data were obtained when colloidal silica was used, a Na-magadiite phase was obtained starting from 2.4 g of NaOH to 11 g (runs 40–42), and no dissolution of solids occurred. The PXRD patterns were similar to the one of Na-magdiite with a $d_{001}$ of 1.54 nm (Figure 4 (left)).

### 2.3. Effect of Water Content

Since Ludox-AS40, Ludox-HS40, and colloidal silica contain water, the fumed silica was chosen to avoid a possible interference of additional water contents. The temperature was fixed at 150 °C for a duration of two days. The amounts of silica and NaOH were also fixed. Indeed, when colloidal silica was used, without adding water to the mixture, meaning the NaOH solid was directly added to the colloidal silica (run 43), a Na-magadiite phase was formed, because the colloidal silica contained about 60% of water.

However, in the case of fumed silica, the PXRD data indicated that Na-magadiite was formed when 50 g of water or more was used in the gel (runs 18, 19, and 2), as revealed by the value of $d_{001}$ of 1.54 nm (Figure 5). However, a mixture of Na-kenyaite and Na-magadiite phases was produced with two values of $d_{001}$ at 2.01 nm and 1.56 nm (runs 17) when 40 g of water was added to the gel. Only pure Na-kenyaite was formed with a smaller mass of water of about 20 g (run 16), and the PXRD pattern exhibited one value of $d_{001}$ of 2.10 nm (Figure 5).

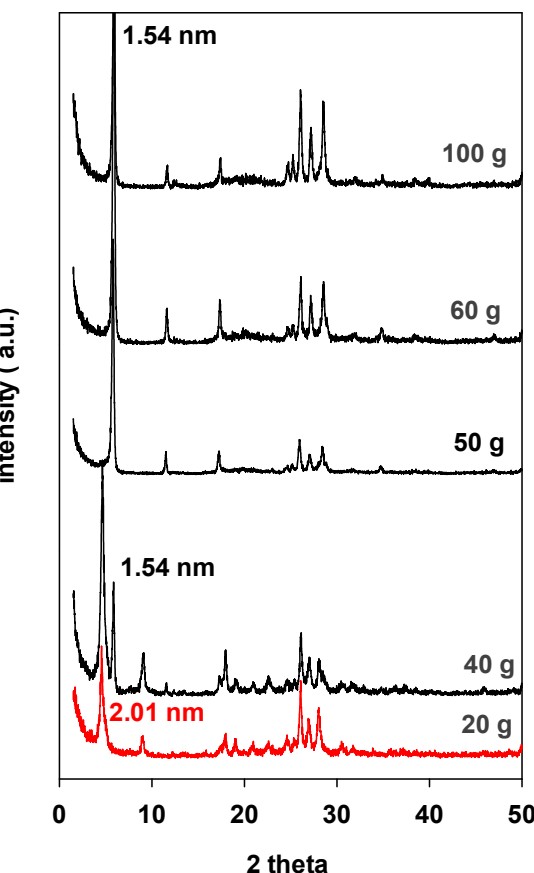

**Figure 5.** PXRD patterns of the samples prepared from fumed silica and using different amounts of water at 150 °C for 2 days.

## 2.4. FTIR Spectra

The FTIR spectra of the Na-magadiite samples prepared from different sources of silica at 150 °C and for two days were collected (runs 2, 21, 25, 30), and a representative spectrum of MAG-FS is presented in Figure 6. Two main regions were observed. The first one was from 4000 $cm^{-1}$ to 2000 $cm^{-1}$, and the second one was from 1350 to 400 $cm^{-1}$.

The OH stretching vibrations ($\nu_{OH}$) of interlamellar adsorbed water as well as hydrogen-bonded hydroxyls were detected at 3660, 3580, and 3470 $cm^{-1}$, respectively [31,32]. The presence of a sharp band at 3660 $cm^{-1}$ was assigned to the free hydroxyl groups [33]. The presence of bands at 1672 and 1630 $cm^{-1}$ was due to the bending deformation of water, which suggested that MAG-FS contained different sites that interact with water, including hydrogen bonding between the free water molecules and Si-OH groups and the hydration water of sodium cations, respectively [34,35].

In the region from 1350 to 400 $cm^{-1}$, MAG-FS exhibited an intense band at 1081 $cm^{-1}$ with shoulders at 1179 and 1242 $cm^{-1}$, which was associated with the stretching vibration modes of $\nu_{as}$ (Si-O-Si) [31,33]. The position of this bond depends on the synthesis method and the type of cations intercalated between the magadiite layers. The bands at 818 and 781 $cm^{-1}$ were assigned to the stretching vibration modes of $\nu_s$ (Si-O-Si). On the other hand, the symmetric and asymmetric bending vibration modes of (Si-O-Si) were located at 621 $cm^{-1}$ and 443 $cm^{-1}$, respectively [14,31].

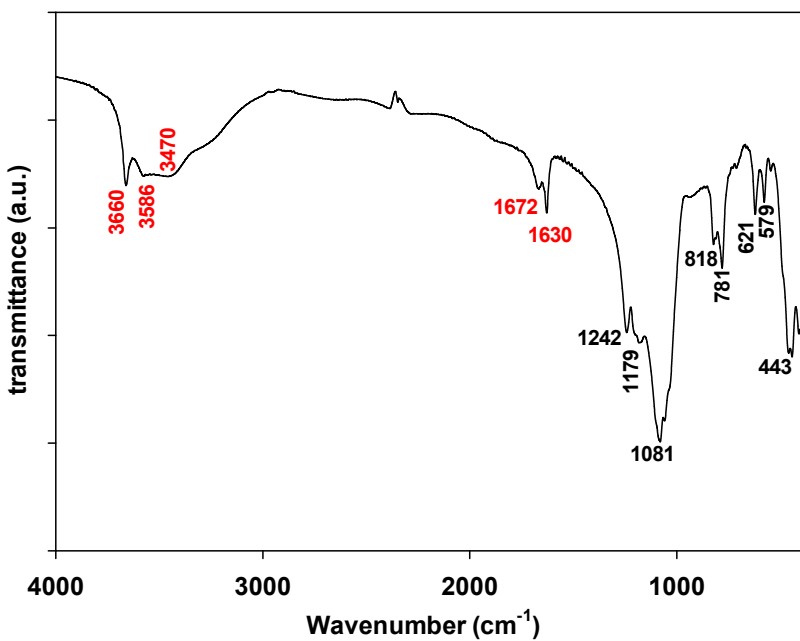

**Figure 6.** FTIR spectrum of MAG-FS (Na-magadiite prepared from fumed silica) material.

The FTIR spectra of materials prepared at different temperatures for a period of two days from fumed silica are presented in Figure 7.

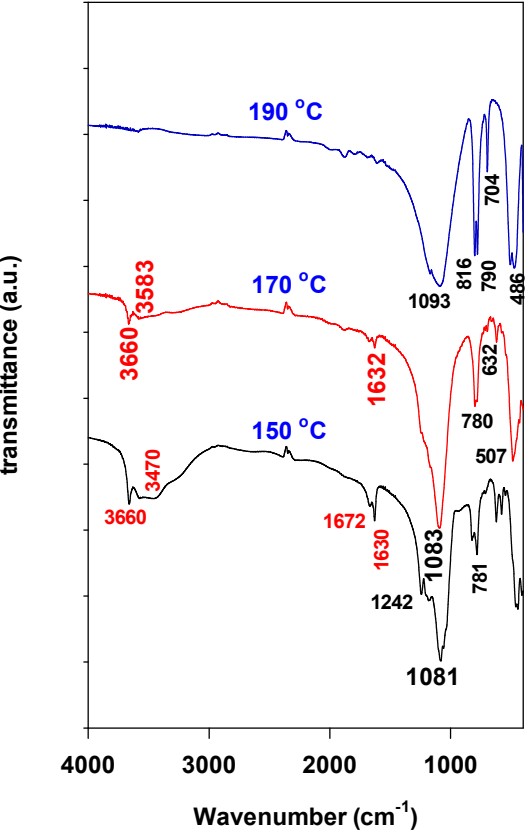

**Figure 7.** FTIR spectra of materials prepared from fumed silica at different temperatures.

A similar spectrum to Na-magadiite was obtained at 150 °C (run 2). At 170 °C, the main feature of the spectrum was preserved; mainly, the band at 3660 cm$^{-1}$ was detected with a shoulder at 3583 cm$^{-1}$ with some changes in the 300 to 800 cm$^{-1}$ region, which

corresponded to a typical Na-kenyaite material (run 8) [33]. As expected, the FTIR spectrum obtained at 190 °C (run 10) only exhibited bands in the range of 1300 to 400 cm$^{-1}$, with a strong one at 1093 cm$^{-1}$; no clear bands were detected below this range, which agreed with the findings reported in the literature for the quartz phase [34].

The FTIR spectra of the samples prepared from fumed silica at 150 °C and for different periods of times are depicted in Figure S3. The FTIR revealed that the spectra were quasi-similar to the Na-magadiite ones (runs 1 and 2). Because the FTIR spectrum of Na-kenyaite was similar to that of Na-magadiite (runs 4, 5, and 6), as shown in Figure 7, it was difficult to identify the bands related to each phase as indicated by the PXRD data (see Figure 2 (right)). At longer reaction periods of 10 days (run 7), fingerprints related to the quartz phase were observed, as shown in Figure 7 (sample at 190 °C).

*2.5. Thermogravimetric Analysis*

The TGA analysis features along with the DTG transformation for the synthesized Na-magadiites prepared from different silica sources at 150 °C for 2 days (runs 2, 21, 25, and 30) are presented in Figure 8.

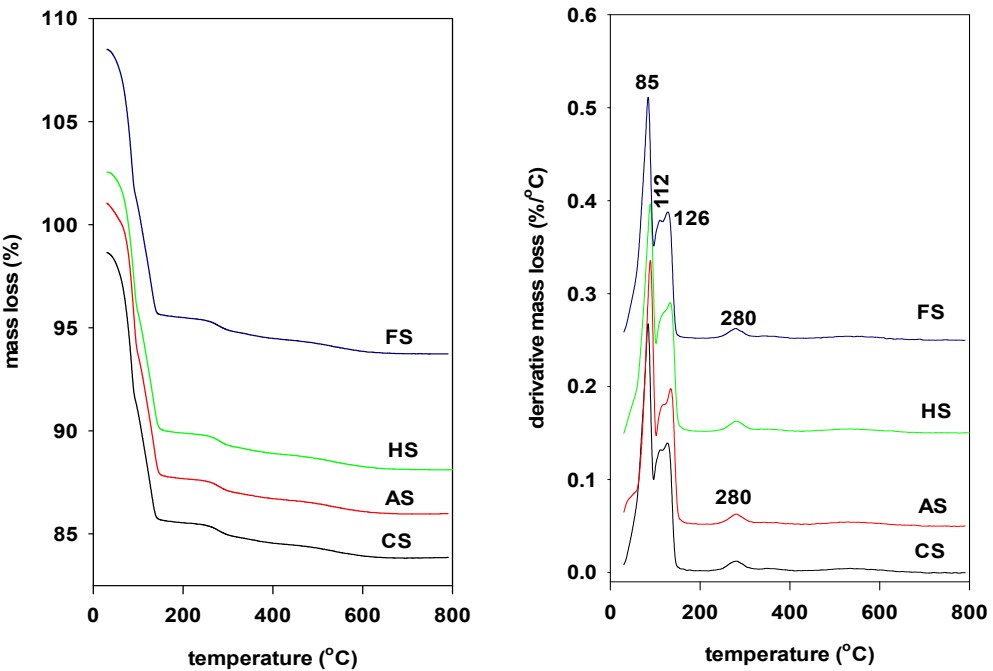

**Figure 8.** TGA (**left**) and DTG (**right**) features of Na-magdiites prepared from different silica sources. CS = colloidal silica, AS = Ludox AS-40%, HS = Ludox HS-40%, and FS = fumed silica.

The Na-magadiites exhibited a mass loss step in the range of 25 to 180 °C (Figure 8, left), as displayed by multiple DTG peaks at 80, 112, and 126 °C, which were attributed to the desorption of different types of water molecules from different environments [11,35] (Figure 8, right). The first step of 5.86% to 7.30% in the range of 25 to 100 °C was assigned to the desorption of the physisorbed water from the surface of the crystals. The second step of 5.57 to 5.74% in the range of 100 °C to 180 °C was associated with the loss of interlayer water molecules and the destruction of the hydration shell of the sodium cations, with two DTG peaks at 112 and 126 °C (Figure 8), respectively [36]. The steady decrease in mass loss of about 0.70% at temperatures above 200 °C was assigned to the framework of silanol dehydroxylation, forming siloxane bonds with a weak DTG peak at 280 °C. A continuous and undefined mass loss of 1% was also observed for temperatures higher than 300 °C until 800 °C, and it was attributed to the complete dehydroxylation of the layered structure [14]; no DTG peaks were recorded in this process. The total mass loss in the range of 25 to 800 °C

was 15 to 14% (Table S1). Overall, the type of silica source did not affect these values, and they were close to those reported for similar materials [37].

Figure 9 (left) presents the TGA features of samples synthesized at different temperatures using fumed silica for a period of 2 days (runs 7, 2, 8, and 10). The features are similar with the three distinct mass loss steps reported above, with a decrease in the mass loss percentages from 14.02 to 8.37% in the range of 25 to 180 °C for the samples prepared at temperatures of 130 to 170 °C (Table S1). These DTG features indicate that Na-magadiite (run 2) and Na-kenyaite (run 8) were similar and in agreement with the findings reported in the literature. Na-magadiite exhibited a higher mass loss percentage of 11.4% compared to Na-kenyaite (8.4%). The TGA and DTG curves were different for the sample prepared at 190 °C and above (runs 10 and 11) due to the formation of only a quartz phase [37], and a single broad DTG signal at 472 °C was detected (Figure 9, right). The total mass loss from 25 °C to 800 °C did not exceed 1.16%, as presented in Table S1.

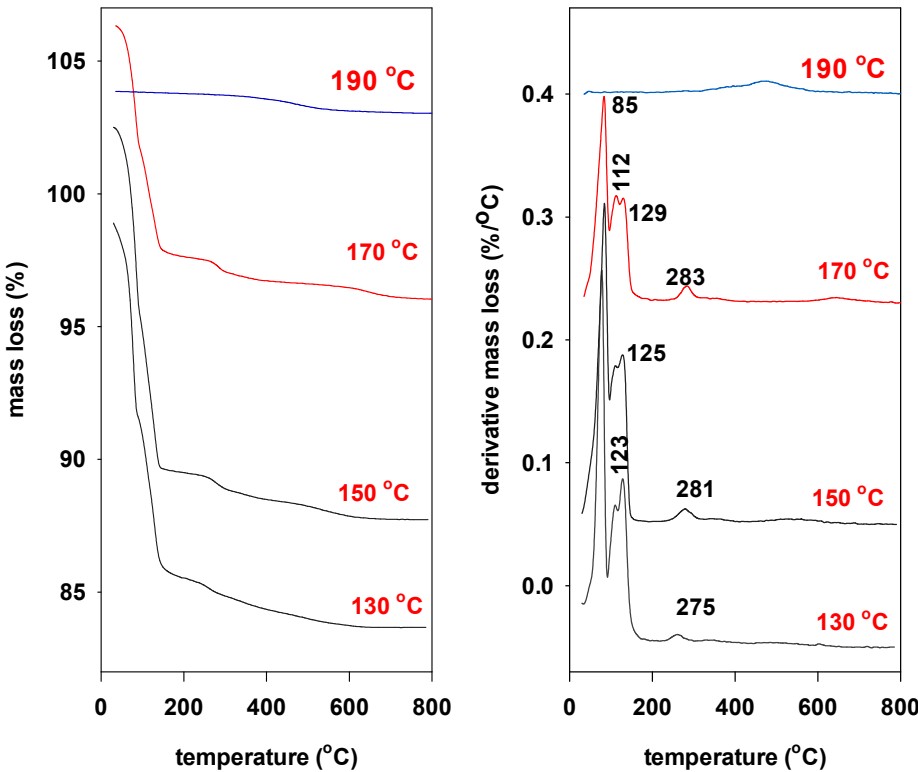

**Figure 9.** TGA (**left**) and DTG (**right**) features of materials prepared from fumed silica at different temperatures for 2 days.

For the samples prepared from fumed silica at 150 °C and at different periods of time (runs 1–6), the TGA and DTG features are presented in Figure S4, and the percentages of mass losses are presented in Table S1. While the general appearances of the TGA features were comparable, differences in the mass loss in the range of 25 to 180 °C were observed and could be related to the nature of phases existing in the resulting samples. It varied from 12.61% to 6.03%. This variation was related to the types of phases present in the product (see Table 1). In some reported data, the differences in the water content and temperatures of water release could be related to the synthesis and measurement conditions [38]. Qualitatively, the percentage mass loss in the range of 25 to 800 °C decreased as the period of synthesis was becoming longer. It varied from 14.44% (1 day) to 7.50% (10 days). The intensity of the DTG peaks at 112 °C and 128 °C decreased in intensity for longer periods of hydrothermal treatment (runs 1–6; Figure S4), and it was assigned to the decrease in the Na-magadiite phase in the products (Table 1).

A typical DTA curve of MAG-FS (run 2) is presented in Figure 10, and it shows that the loss of different types of water molecules is accompanied by endothermic peaks at 91 °C, 119 °C, and 139 °C. At higher temperatures, a weak endothermic peak was detected at 692 °C, and an exothermic peak was observed at 740 °C due to the complete dehydroxylation of the magadiite phase and the recrystallization of the amorphous silica phase [38]. The position of this peak shifted at higher temperatures (from 740 to 775 °C) due to the presence of a silica phase (quartz) in the starting material and due to the presence of a Na-kenyaite phase.

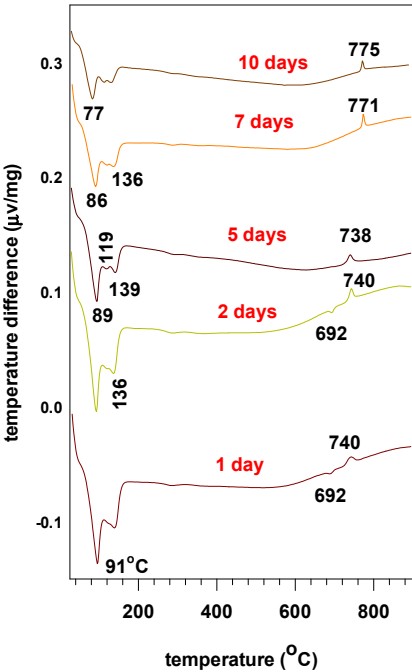

**Figure 10.** DTA curves of materials prepared from fumed silica at 150 °C for different reaction times.

Qualitatively, the intensity of the endothermic peaks decreased at longer reaction times (runs 3–6) and was related to the crystalline phases produced and their types (Figure 10).

The in situ XRD study revealed that crystalline MAG-AS (run 25) was stable up to 400 °C, with a decrease in the basal spacing from 1.54 to 1.38 nm when heated at 100 °C, and it continued to dwindle to a value of 1.15 nm when heated at 150 °C; this was maintained up to 420 °C. A value of 1.15 nm is close to that ascribed for completely dehydrated magadiite [38]. In some cases, a basal spacing of 1.54 nm was maintained even at 240 °C and 400 °C [28,39]. The temperature limit of the in situ device used is 425 °C, so the MAG-AS was calcined at 500 °C, 740 °C, and 820 °C in a furnace under the same conditions as the TGA experiments. The obtained PXRD patterns are presented in Figure S5. The layered structure was completely transformed to an amorphous silica phase at 500 °C. The recrystallization of this amorphous phase to quartz occurred at a temperature of 820 °C, as presented in Figure S6. Different data were reported for the in situ XRD Na-magadiite for temperatures higher than 800 °C, where the layered structure was thermally stable up to 800 °C [2].

### 2.6. $^{29}Si$ MAS NMR Data

The $^{29}$Si MAS NMR offers an additional method to characterize the obtained silicate layered materials. Different runs were performed to distinguish between the different environments of the Si species and to estimate the ratio of $Q^4$ Si species to the $Q^3$ Si ones. The $^{29}$Si MAS NMR spectra of the Na-magadiite samples prepared from the four silica sources at a temperature of 150 °C and for two days (runs 2, 21, 25, and 30) are presented in Figure 11 (left).

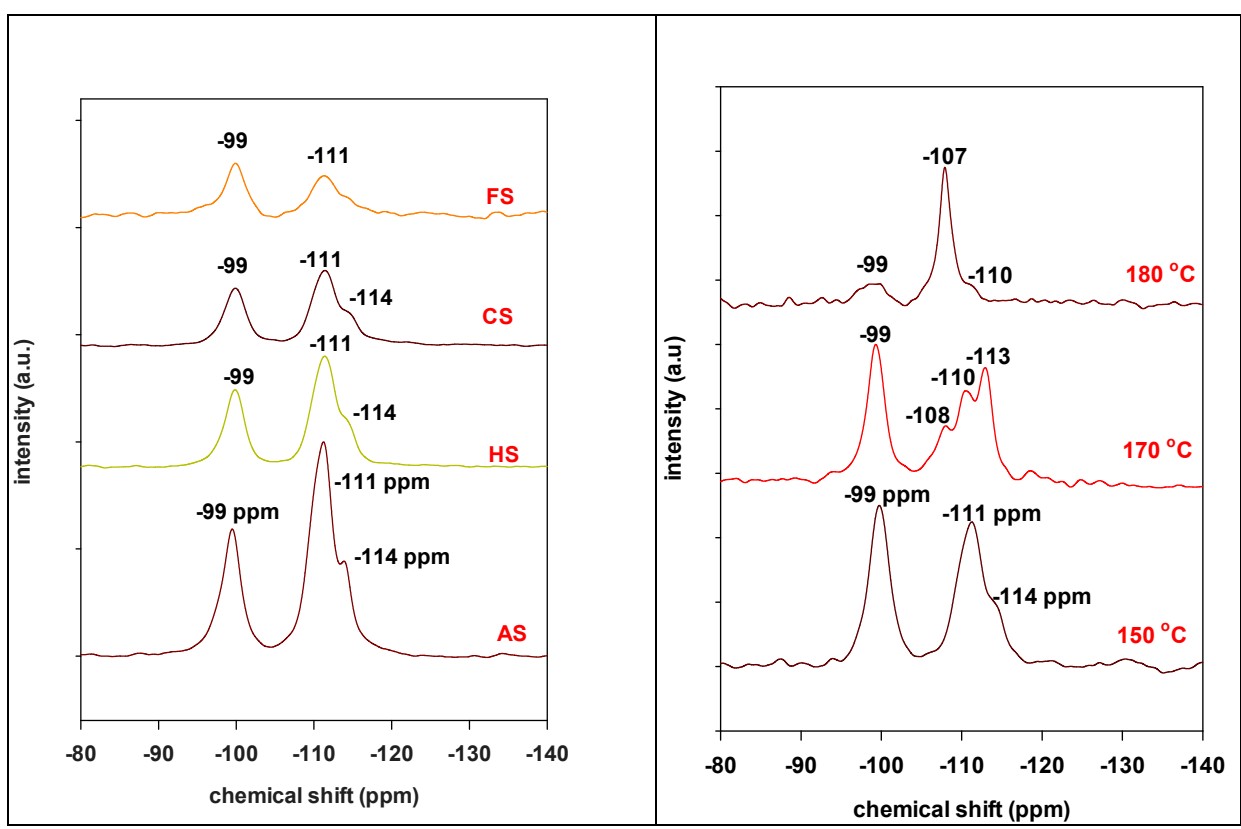

**Figure 11.** $^{29}$Si MAS NMR of Na-magadiite prepared from different silica sources at 150 °C for 2 days (**left**). The **right** side corresponds to materials prepared at different temperatures for a period of 2 days.

The spectra are similar and exhibit one main resonance peak centered at −99 ppm, which is attributed to the chemical structure of the Q$^3$ [Si(3OSi)(O$^-$)] and/or [Si(3OSi)(OH)] sites and due to the extended H-bonded silanols. At least two different Q$^4$ (Si(OSi)$_4$) silicon sites at −111 and −114 ppm were also recorded [33,36]. Generally, the intensity of the Q$^4$ Si species was higher compared to the Q$^3$ Si species except for MAG-FS (run 2).

To compare the actual ratios of Q$^4$ to Q$^3$ intensity to the published values, one has to be careful to ensure that the synthesis and data collection conditions are the same. The Q$^4$-to-Q$^3$ theoretical ratio was 2.5 [26]; however, the experimental values varied from 1 to 5 [40–43]. In the present study, the ratios varied from 0.9 to 2.25, and they depended on the silica sources; the Q$^4$-to-Q$^3$ ratio was 2.25 for MAG-AS, 1.79 for MAG-HS, 1.59 for MAG-CS, and 0.9 for MAG-FS (Figure S6).

The $^{29}$Si MAS NMR spectra of the samples prepared at different temperatures from fumed silica (runs 2, 8, and 10) are presented in Figure 11 (right). All of the spectra exhibited a peak at −99 ppm, originating from the silicon Q$^3$ sites, with a reduced intensity for the samples prepared at 190 °C (run 10). The latter showed one strong peak at −108 ppm, indicating the presence of the quartz phase as the main phase. However, for the sample prepared at 170 °C, a clear triplet of silicon Q$^4$ sites was observed at −108 ppm, −110 ppm, and −113 ppm (run 8). The overlap of these bands was achieved at 150 °C with a main peak at −111 ppm and a shoulder at −114 ppm (run 2). The Q$^4$-to-Q$^3$ ratio was estimated to be in the range of 1.26 for Na-magadiite and 1.18 for Na-kenyaite.

The $^{29}$Si MAS NMR spectra of materials prepared from Ludox AS40 at 150 °C for different reaction times (runs 24–28) are presented in Figure S7. The Ludox-AS 40 was selected as a silica source because the Na-magadiite phase was obtained in a major phase after 5 days of hydrothermal treatment (run 27). Next, we investigated if there was a change in the Q$^4$-to-Q$^3$ ratio with the increasing synthesis time. The spectra were similar for the obtained materials from 1 to 7 days and consisted of a Na-magadiite phase with two main

resonance peaks at −99 ppm and −111 ppm, respectively, with a shoulder at −114 ppm (runs 14–27). However, after 10 days of hydrothermal treatment (run 28), the intensity of the two peaks at −99 ppm and −111 ppm decreased, and a new peak appeared at −108 ppm; this peak could be related to the silica phase present in the sample (see Figure S7). The area of the $Q^3$ Si site peak also decreased due to the partial transformation of Na-magadiite to Na-kenyaite. The $Q^4$-to-$Q^3$ ratios changed from 2.1 to 2.80.

$^{29}$Si MAS NMR was also performed for the samples prepared using different amounts of water and fumed silica as a source of silica (runs 16–19) (not shown). The spectra were similar in shape to the silicon $Q^3$ type at −99 ppm, and a wide peak centered at −111 ppm was conserved with some shoulders in the range of −104 to −120 ppm. These data indicated that the different types of $Q^4$ silicon species also depended on the amount of water used in the synthesis. The $Q^4$-to-$Q^3$ ratios were in the range of 1.00 to 1.74.

*2.7. SEM-EDX Analysis*

The EDX analysis indicated that three main elements existed in all of the samples (O, Na, and Si). The content of Na was in the range of 3.77 to 5.76%, which is close to that reported in the literature (Figure 12). These values depended on the synthesis conditions and mainly on the starting $Na_2O/SiO_2$ ratios. The samples prepared at a temperature of 185 °C and above (runs 9–11) contained a lower percentage of Na close to zero, which resulted in the formation of only a silica phase.

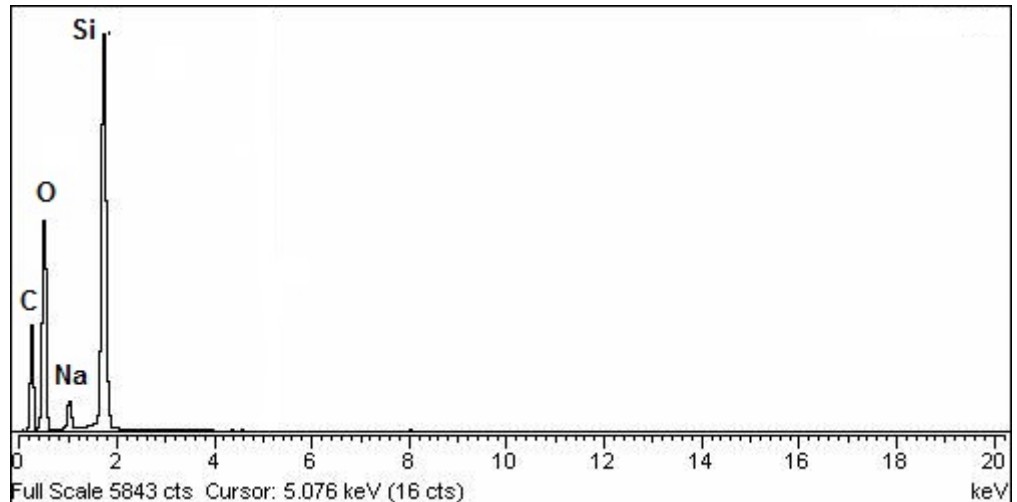

**Figure 12.** Energy-dispersive X-ray (EDX) analysis of MAG- (CS).

The SEM micrographs of Na-magadiites prepared at 150 °C for 2 days and using different sources of silica (runs 2, 21, 25, and 30) are presented in Figure 13.

Overall, the morphologies of the Na-magadiites prepared from silica precursors in suspension (Ludox HS-40%, Ludox AS-40%, and colloidal silica) were similar and agreed with those reported in the literature. The shape consisted of spherical nodules resembling rosettes [42]. Other authors described the morphology of Na-magadiite samples as a typical cauliflower morphology with an open aggregation of plates [43], while a study described Na-magadiite as having a petal-like lamellar structure [11]. However, for the Na-magadiite prepared from a solid silica source (fumed silica; run 2), the morphology was quite different: a rosette shape was not clearly observed, and the particles were aggregated like plates.

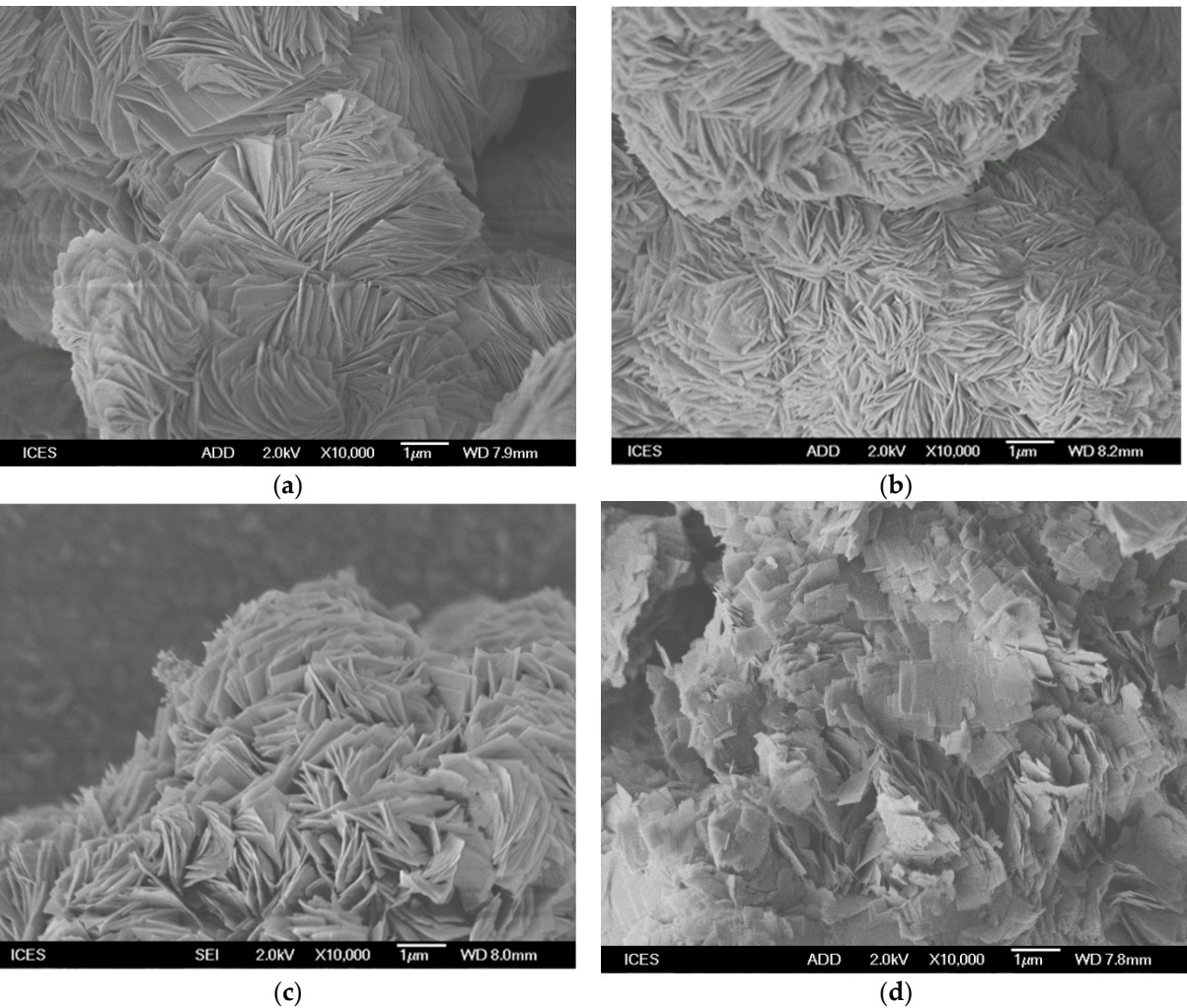

**Figure 13.** SEM micrographs of (**a**) Mag-AS, (**b**) MAG-HS, (**c**) MAG-CS, and (**d**) MAG-FS.

Interestingly, the rosette-like structure was obtained from fumed silica when the gel was treated for 2 days at 130 °C (run 7, Figure 14a). This structure was lost when the gel was treated at 150 °C or 170 °C for 2 days (runs 2, 8), as reported above. The Na-kenyaite phase was obtained instead of Na-magadiite (run 8, Figure 14b). At the higher temperature of 190 °C (runs 10 and 11), the resulting silica quartz phase exhibited irregular hexagonal shapes, as shown in Figure 14c.

The changes in the morphology were examined for Na-magadiite prepared from Ludox-AS40% as the silica source for different reaction times at 150 °C. The rosette-like structure was maintained for a period of hydrothermal treatment for up to three days (runs 24–26). At 5 days (run 27), the rosette morphology was lost, and the shape of the particles changed, as presented in Figure S8. The PXRD data indicated that the type of phase did not change, and mainly Na-magadiite was obtained with traces of kenyaite and silica phases (run 28). However, when using fumed silica and different reaction times at 150 °C, the rosette-like structure was not observed even after one day of reaction (run 1; see Figure S9).

The effect of water content on the morphology of Na-magadiite was also investigated using fumed silica. As reported above, adding 20 g to the gel composition (run 16) led to the synthesis of Na-kenyaite with stacked plates and with different shapes. In the case of a mixture of Na-magadiite and Na-kenyaite, the morphology was the same as that of the sample of pure Na-kenyaite (run 17). However, when 60 g of water was used (run 19), the morphology of the plates was changed with regular rectangular and square shapes (Figure 14d); a twisted morphology of these plates was achieved when a higher content of

water was used (about 105 g; run 2). In the last two cases, only the Na-magadiite phase was detected via PXRD.

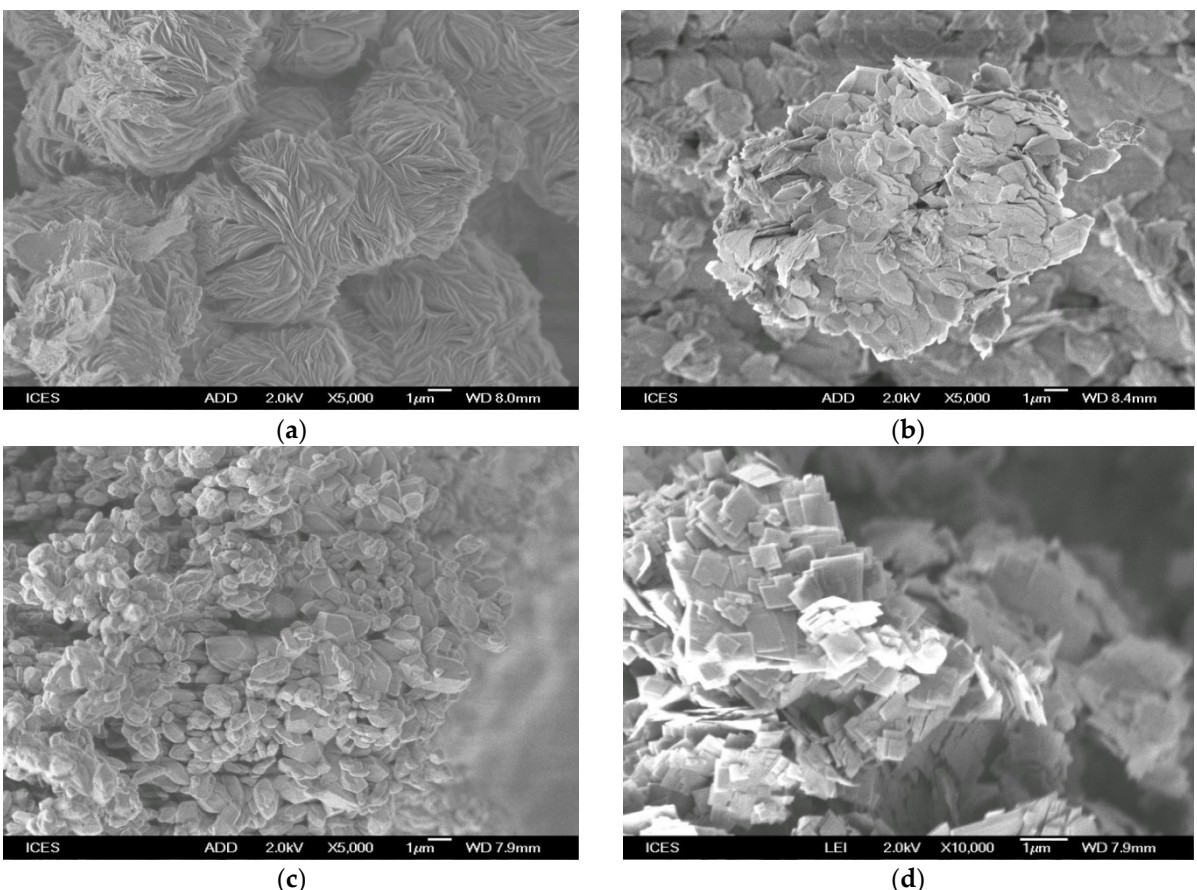

**Figure 14.** SEM micrographs of samples prepared from fumed silica at different temperatures: (**a**) 130 °C, (**b**) 170 °C, and (**c**) 190 °C. (**d**) Na-magadiite prepared at 150 °C for 2 days using 60 g of water.

### 2.8. Surface Properties

The surface properties of selected pure Na-magadiites are presented in Table 3. The Na-magadiite materials exhibited specific surface area values in the range of 30 to 40 m/g. These values had the same magnitude as those reported for similar materials. The total pore volumes were estimated from the adsorption isotherms at a relative pressure of 0.95; the values were in the range of 0.183 cc/g to 0.263 cc/g. The average pore diameters were associated with the voids between the particles and were in the mesopore range. The diameter of the voids depended on the morphology of the particles and the silica source.

**Table 3.** Surface properties of selected pure Na-magadiite samples obtained at different conditions.

| Sample | $S_{BET}$ (m²/g) | T.P.V. (cc/g) | A.P.D (nm) |
|---|---|---|---|
| MAG-AS | 35 | 0.183 | 20.7 |
| MAG-HS | 38 | 0.230 | 23.9 |
| MAG-CS | 40 | 0.263 | 25.8 |
| MAG-FS | 33 | 0.221 | 26.7 |
| MAG-FS (130 °C) * | 42 | 0.211 | 20.1 |
| MAG-FS(60 g) [+] | 29 | 0.202 | 27.4 |

* MAG-FS prepared at 130 °C for 2 days. [+] MAG-FS prepared at 150 °C for 2 days using 60 g of water.

## 3. Removal Properties of Na-Magadiite Materials

In this study, only pure Na-magadiite samples were used, and they were prepared from different silica sources (MAG-As, MAG-HS, MAG-CS, and MAG-FS) and with different morphologies for MAG-FS.

### 3.1. Effect of Initial Concentration of BB-41

In this paragraph, 0.1g of selected MAG-AS was added to 10 mL of BB-41 solutions at various concentrations (25 to 1000 mg/L).

Figure 15 (left) indicates that the removal capacity of MAG-AS increased from 25 mg/g to a maximum of 190 mg/g at a $C_i$ value of 250 mg/L, and then it remained almost constant with a slight increase to 200 mg/g above this initial concentration. It was reported that the initial concentration acts as a driving force to overcome the mass transfer resistance of the cationic dyes between the aqueous and solid phases [44,45]. However, by increasing the $C_i$ values, the removal percentage of BB-41 decreased, as presented in Figure 15 (left). This fact could indicate for a fixed Na-magadiite dose and a fixed volume of BB-41 that the restricted accessible removal sites might play major roles in the decrease in the removal percentage at higher initial dye concentrations. Similar trends were observed for other solids and for the same dye or for different dyes [45–47].

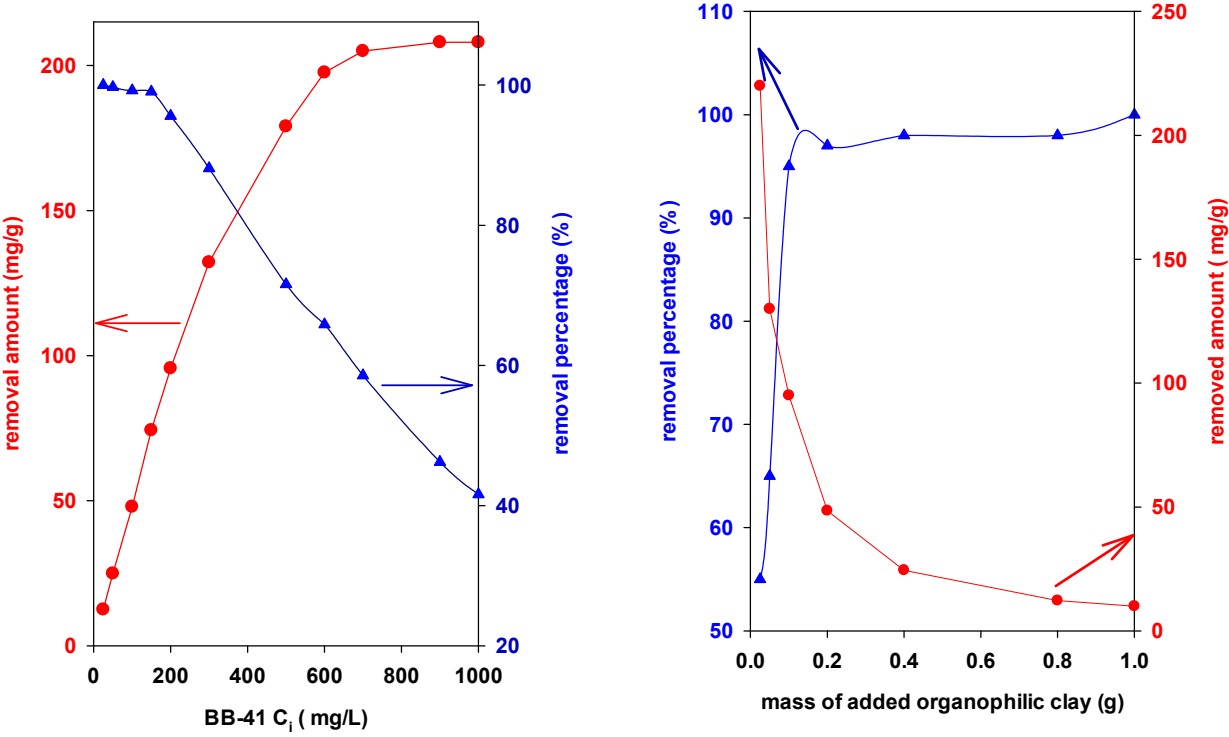

**Figure 15.** Influence of initial concentration of BB-41 (**left**) and the mass of added MAG-AS on the removal of BB-41 (**right**).

### 3.2. Effect of Na-Magadiite Dose

The initial concentration of BB-41 was fixed at 200 mg/L which was the concentration at which the maximum removed amount was obtained. The results are presented in Figure 15 (right). The percentage of removal was enhanced with the addition of more MAG-AS, increasing from 55% to 95% for added masses of 25 to 100 mg of MAG-AS; above this amount, only a slight improvement was observed, and it reached a maximum of 100% when 1 g was added. Meanwhile, the removed amount decreased with the increasing amount of MAG-AS.

Next, the volume of the solution was fixed so that the number of BB-41 cations did not change. When the mass of MAG-AS was increased, a larger number of removal sites was available, and they were fully occupied by the cationic dyes, leading to an increase in the removal percentage (R%). On the other hand, for a lower mass of MAG-AS, there were fewer removal sites, and they were enough to remove all of the dye molecules. In addition, Equation (S2) indicates that the removal amount ($q_e$) was inversely proportional to the mass of adsorbent. Thus, the maximum removed amount of BB-41 occurred at lower masses of MAG-AS.

### 3.3. Effect of pH

The acidity of the dye solution is considered to be an important factor for the removal process; it affects the types of the species that exist in the solution and the charge of the used solid [48]. The pH of the BB-41 solution was tuned by adding 01M HCl or NaOH solution, separately. The data indicated that there was an effect of the solution pH on the removal percentage of BB-41 by the MAG-AS sample. The removal percentage increased from 45% (pH = 2) to 95% (pH = 10) (Figure 16, left). Above this pH, at about 10.5, a brown precipitate was formed, and no removal measurement was performed [44].

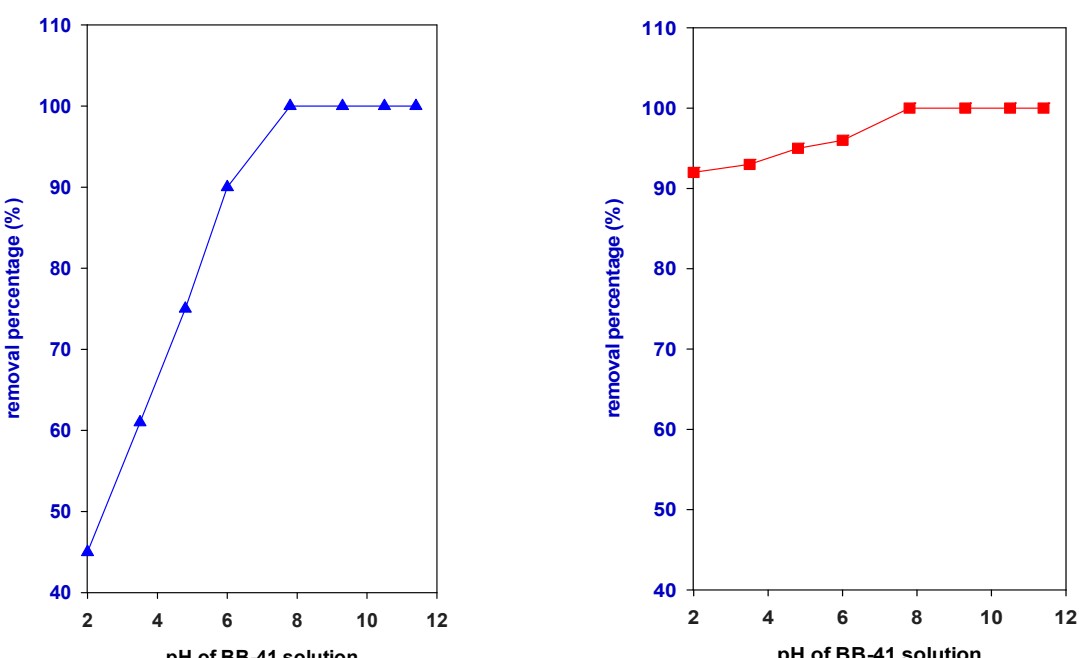

**Figure 16.** Effect of pH on BB-41 solution using (**left**) MAG-AS washed until a pH of 7 and (**right**) MAG-AS without controlling the pH after washing.

The convenient pH range was from 8 to 10.00. The pH values also affected the charge of the Na-magadiite; indeed, the $PH_{PZC}$ reported for Na-magadiite was close to 4.5 to 5.1 [12,47], another value of 8 was also mentioned [35]. At pH values higher than $pH_{PZC}$, the MAG-AS exhibited a negative surface charge. The higher removal percentage of BB-41 dye was explained by the electrostatic interactions between the surface charge of the MAG-AS and the positively charged dye cations. In the case of lower pH values, the surface exhibited a positive charge, and a competition between the protons' ions and the positively charged dyes occurred, resulting in lower percentage removal values. Different data were obtained for MAG-AS after washing with distilled water without controlling the final pH. In this case, a slight change was observed from 92 to 96% for a pH range from 2 to 8. Then, 100% of removal was attained at pH values greater than 8 (Figure 16, right). This fact can be related to the high basic character of MAG-AS water suspension. Indeed, the

suspension of MAG-AS and water exhibited a pH value above 10, close to that reported in the literature [49].

When added to the BB-41 solutions with acidic pH values, the final pH of the suspension was between 7.5 and 8, and thus affected the percentage of BB-41 removal. For example, when using another basic dye (methylene blue), the removal percentage increased with the increase in the pH, and the best pH interval for MB adsorption on Na-magadiite was from 8.5 to 10.0 [12]. In this case, the experimental methods and the conditions need to be optimized and unified for comparison purposes.

### 3.4. Effect of Silica Sources

If the pH of the suspension was closely controlled during the washing process, MAG-CS, MAG-AS, and MAG-HS removed about the same amounts of BB-41 at 1000 mg/L; however, for MAG-FS, less BB-41 was removed, and a decrease of 20% was achieved for initial concentrations higher than 300 mg/L. The variations in BB-41 uptake could be associated with the surface areas of the Na-magadiites or with the morphology of the Na-magadiite particles. Indeed, MAG-AS, MAG-HS, and MAG-CS exhibited rosette-like structures, which was not the case for the Na-magadiite obtained from solid fumed silica. The content of $Na^+$ cations in the Na-magadiite samples that determine the CEC values could contribute to these results. Indeed, a lower percentage of $Na^+$ cations was detected for the MAG-FS sample.

### 3.5. Effect of Morphology Shape

Next, we studied three different shapes of pure Na-magadiite prepared from fumed silica including cauliflower- or rosette-like shapes (130 °C; 2 days), loose discrete rectangle plates with inhomogeneous sizes (60 g of water at 150 °C for 2 days), and abnormal cauliflower shapes (150 °C for 2 days with 105 g of water) with a plate-like morphology, although the plates were quite irregular and some had an edge-curved shape.

The removal tests indicated that the maximum amount of BB-41 removal (217 mg/g) was obtained by the rosette-like morphology. However, the Na-magadiite with loose discrete rectangle plates exhibited the lowest value of 150 mg/g. An intermediate value of 167 mg/g was obtained using Na-magadiite with a plate-like morphology.

### 3.6. Langmuir Adsorption Models

The Langmuir model was applied to the experimental data [50]. The search for the best fit adsorption isotherm using the linear regression method is widely used to evaluate the model parameters and to determine the best fitting model. The method of non-linear regression is used by several researchers to determine the adsorption isotherm parameters. This method is sometimes used to avoid errors affecting $R^2$ during linearity [51]. Our aim was not to compare the two methods, but to estimate the maximum amount of BB-41 that can be removed by the Na-magadiite materials. The Langmuir model assumes uniform energies of adsorption onto the surface and no transmigration of adsorbate in the plane of the surface [50]. The Langmuir parameters are defined as follows: $q_{max}$ is the maximum adsorption capacity (mg/g) and $K_L$ is the constant related to the free energy of adsorption (L/mg). The corresponding parameters for both models are presented in Table 4. First, pure Na-magadiite materials prepared from colloidal silica forms exhibited the highest maximum removal capacity, and the one from solid fumed silica had a lower removal capacity of 167 mg/g. The morphology of the materials could be one of the reasons, allowing for more access to the removal sites. The surface area could be an additional parameter; however, the magadiite samples with a rosette-like structure exhibited an average value of 37 $m^2$/g, while MAG-FS displayed a surface area of 33 $m^2$/g. This slight variation could not be the origin of this difference.

**Table 4.** Langmuir model parameters for the removal of BB-41 for different materials using non-linear and linear equations.

| Samples | $q_{max}$ (mg/g) | $K_L$ (L/g) | $R^2$ |
|---|---|---|---|
| MAG-AS | 222 (219) * | 0.068 (0.112) | 0.9172 (0.9998) |
| MAG-HS | 212 (210) | 0.065 (0.098) | 0.831 (0.9985) |
| MAG-CS | 205 (208) | 0.065 (0.112) | 0.9491 (0.9978) |
| MAG-FS | 172 (167) | (0.061) (0.064) | 0.9386 (0.9889) |
| MAG-FS (130 °C) | 220 217 | 0.071 (0.142) | 0.9453 (0.9857) |
| MAG-FS (60 g) | 153 (150) | 0.063 (0.066) | 0.9406 (0.9987) |

* Values between brackets correspond to the linear equation.

To improve the removal capacity of Na-magadiite prepared from fumed silica, the synthesis conditions were slightly changed. Using a synthesis temperature of 130 °C for 2 days, the rosette-like morphology was obtained, and the removal capacity was increased to 217 mg/g. For another sample, which was prepared at 150 °C for two days, this morphology was lost, resulting in a decrease in the removal capacity to 167 mg/g. The shape of the plates affected the maximum removal capacity, and it decreased from 210 mg/g to 150 mg/g, as presented in Table 4.

Table 5 compares the maximum removal amounts of BB-41 by the prepared magadiites with other materials reported in the literature as removal agents for the same dye. The reported data in Table 5 were estimated using the Langmuir linear equation. The use of this form was necessary for comparison purposes. The data indicated that the materials synthesized in this work have great potential as removal agents for this cationic dye from aqueous media. The compared materials were limited to layered materials and aluminosilicates. The maximum removal capacity of 219 mg/g was higher than that of several adsorbent materials published in the literature, from solid waste as well as from mineral sources.

**Table 5.** Removal capacities of different materials for BB-41 dye.

| Material | Removal Capacity (mg/g) | Reference |
|---|---|---|
| Na-magadiites | 150–220 | This work |
| Sol gel silica material from grape bagasse | 268 | [52] |
| Saudi local clay mineral | 74 | [53] |
| Zeolite tuff | 192 | [54] |
| Brick waste materials | 60–70 | [46] |
| Nanoporous silica | 345 | [55] |
| Mn-modified diatomite | 62 | [56] |
| Clinoptilolite//$Fe_2O_3$ nanoparticles | 93 | [57] |
| Natural Gordes zeolite | 149 | [58] |

*3.7. Regeneration Data*

The dyes were moved from the aqueous phase to the surface of the adsorbent agents using an adsorption process. The spent agent may turn into a possible pollutant depending on the adsorbate and cannot, therefore, be dumped in a landfill [59]. Regenerating and reusing the adsorbent is crucial. The three main categories of conventional procedures for regeneration are physical, chemical, and biological processes [60,61]. In a different instance, the thermal desorption approach was suggested, where the spent adsorbents were heated at temperatures higher than 100 °C [61]. Oxidation, which is utilized in chemical processes and is known as the Fenton process, is thought to be effective at degrading organic adsorbates

and is also the most affordable and common choice [62]. Nevertheless, one drawback of many regeneration techniques is that the by-products they produce might be harmful. As reported in Section 2.4, the degradation of the removed BB-41 was adopted using oxone and cobalt nitrate solutions [25]. This procedure was simple to operate and led to a good performance. An initial BB-41 concentration of 200 mg/L was selected, and MAG-HS and MAG-FS with removal percentages of 95% and 57%, respectively, were investigated.

The data indicated that MAG-FS maintained its efficiency after four cycles with a slight decrease of about 12% from the initial value, and then it continued to decline to 20% after the fifth cycle. After the seventh cycle, the efficacy was reduced to 32% of the starting value (Figure 17).

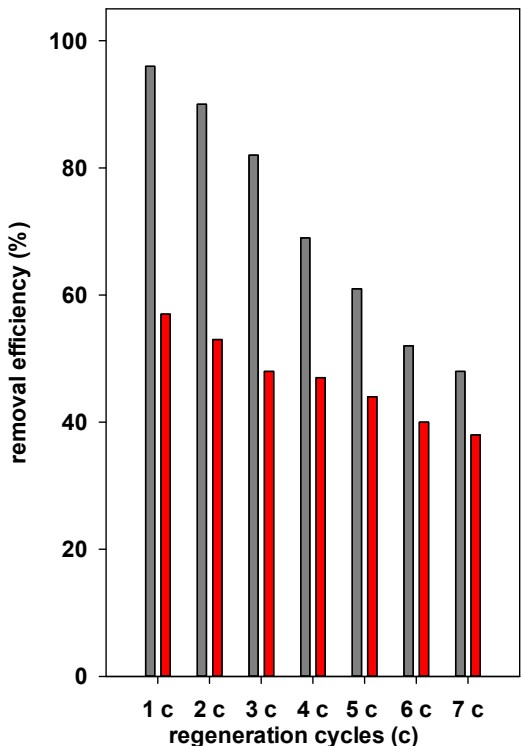

**Figure 17.** Regeneration tests of MAG-AS (grey) and MAG-FS (red).

In the case of MAG-HS, the regeneration experiments indicated that this material lost about of 20% of its efficacy after the first two cycles, and the percentage removal was reduced gradually after the third run, varying from 60 to 50%. However, 50% of efficacy was still preserved after the seventh run (Figure 17). The difference between the two materials could be related to the high removal capacity of MAG-HS, thus making it difficult to destroy all of the removed BB-41 cations. Another reason could be related to the morphology of the rosette-like structure that makes it difficult to access the removed BB-41 cations. The same method was used to regenerate a porous clay heterostructure with a high surface area and a mesoporous character, and similar data were obtained. In this case, the accessibility of the catalyst to the BB-41 located in the pores allowed for their degradation and maintained the removal efficiency [62].

The PXRD runs of the reused magadiites after each cycle indicated that the structure was slightly altered, and the overall pattern corresponded to pure Na-magadiite. These data confirm that Na-magdiitte was chemically stable at the reported conditions.

### 3.8. Batch Adsorber Design

Next, we proposed a single-batch design using two MAG-HS and MAG-FS samples. The design model was used to estimate the predicted masses necessary to reduce a fixed volume of a specific initial BB-41 concentration ($C_i$, mg/L) to targeted $C_1$ (mg/L) [63]. The

amount of adsorbent is $M$ (g), and the solute loading changes from $q_0$ to $q_1$ (mg/g). At the time $t = 0$, $q_0 = 0$, and as time proceeds, the mass balance links the BB-41 removed from the liquid to that picked up by the solid, as reported in Equation (1) [64]:

$$V(C_0 - C_1) = M(q_0 - q_1) = Mq_1 \tag{1}$$

At equilibrium, $C_1 = C_e$ and $q_1 = q_e$, and the equation can be transformed to Equation (2):

$$\frac{M}{V} = \frac{C_0 - C_e}{q_e} \tag{2}$$

Since the Langmuir model fits the data well, the $q_e$ can be substituted, and it is converted to Equation (3):

$$\frac{M}{V} = \frac{C_0 - C_e}{q_e} = \frac{C_0 - C_e}{\frac{q_m K_L C_e}{1 + K_L C_e}} \tag{3}$$

Knowing the intended percentage of reduction (R%) in the initial concentration and the fixed value of $C_i$, the concentration at equilibrium $C_e$ can be easily deduced.

Figure 18 (left) depicts the relationship between the predicted masses of MAG-AS needed to reduce an initial BB-41 concentration of 200 mg/L to different concentrations and using different volumes (V) of BB-41 solutions at room temperature.

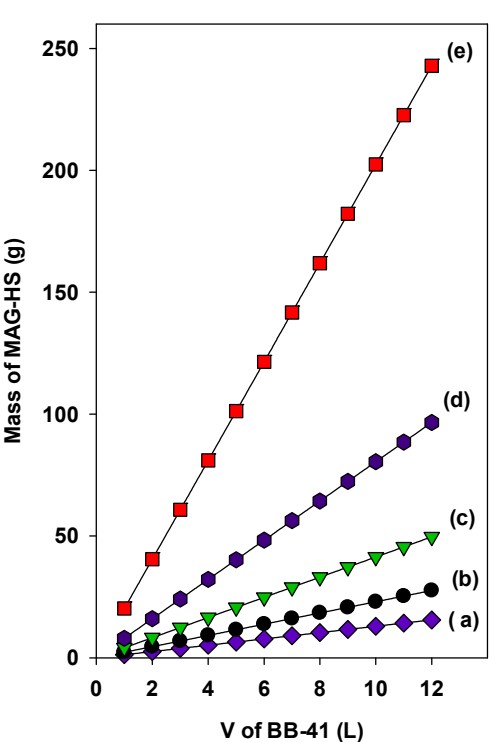 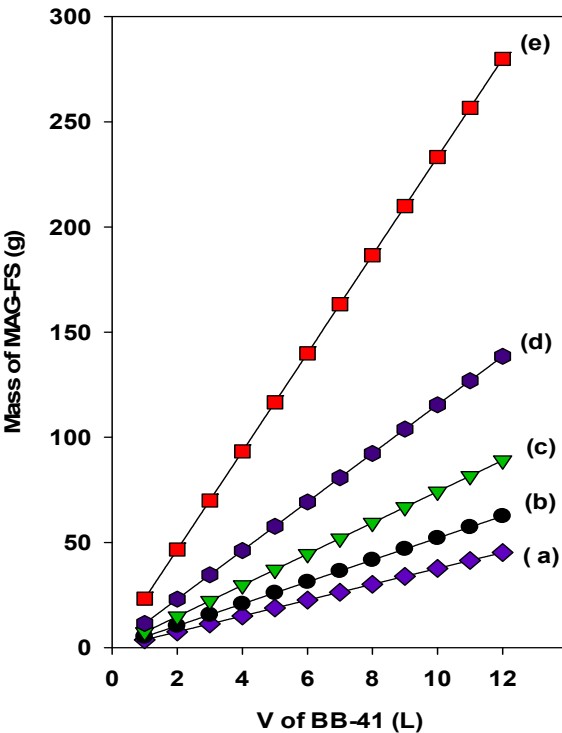

**Figure 18.** The predicted masses (g) of MAG-HS (**left**) and MAG-FS (**right**) required to reduce different volumes (L) of BB-41 (Ci = 200 mg/L) to (**a**) 50%, (**b**) 60%, (**c**) 70%, (**d**) 80%, and (**e**) 90%.

Two general trends were observed: the mass of MAG-AS or MAG-FS kept increasing with the increasing volume and the removal percentage. Indeed, for a fixed volume of 10 L, the required masses of MAG-AS were estimated to be 13 g, 23 g, 41 g, 80 g, and 202 g to achieve the target concentrations of 100 mg/L, 80 mg/L, 60 mg/L, 40 mg/L, and 20 mg/L, respectively.

Meanwhile, when MAG-FS was used, masses of 38 g, 52 g, 74 g, 115 g, and 237 g were needed to obtain removal percentages of 50%, 60%, 70%, 80%, and 90% for 10 L of the BB-41 solution, respectively (Figure 18, right).

Generally, the required amounts of MAG-AS were less than those estimated for MAG-FS; this difference was due to the higher maximum removal capacity of MAG-AS compared to the MAG-FS samples. Similar trends were also obtained with different materials and for the same or different pollutant dye [62,63].

The presented figure could be extended to include other values of BB-41 percentage removals until they reached 100%, and they could be modified to include any other conditions from the initial concentration to the solution temperature [63,65].

## 4. Experimental Procedure and Characterization

### 4.1. Materials

Four silica sources (fumed silica (FS), colloidal silica (CS), Ludox AS-40% (AS), and Ludox HS-40% (HS)) and sodium hydroxide were acquired from Aldrich (St Louis, MO, USA). The cationic BB-41 dye, oxone, and cobalt nitrate salt were supplied by Across Organics (Loughborough, UK). All of the chemicals were utilized as received.

### 4.2. Synthesis of Layered Silicate

A typical mixture was prepared by mixing the reagents in the following order: 4.8 g of NaOH was dissolved in 105 g of distilled water, and 45 g of Ludox HS-40% colloidal silica was added drop-wise to the sodium hydroxide solution with stirring for over 30 min [20]. The resulting mixture had a molar composition of $Na_2O/5SiO_2/122H_2O$, and it was further stirred for another 1 h at room temperature. Finally, it was transferred into a Teflon liner autoclave at 150 °C for 48 h in a static oven. After that, the autoclave was quenched immediately in an ice bath, and the sample was separated via filtration and washed with distilled water until a pH close to 7 was obtained, and then it was air dried at 40 °C overnight.

The pure Na-magadiite phase will be identified as MAG(X), where X stands for the silica source. For example, MAG(CS) means Na-magadiite prepared from colloidal silica as silica source.

### 4.3. Removal of Basic-Blue 41 Procedure

The removal of BB-41 was achieved in batch procedure, as reported in previous work [25]. A stock solution of 1000 mg BB-41 was prepared in a 1000 mL volumetric flask, from which a series of dilutions were prepared (25–1000 mg/L). Identical masses of MAG(X) (0.100 ± 0.005 g) were added to 50 mL sealed glass tubes. Obtained suspensions were retained in an automatic shaker at 25 °C overnight at 125 rpm to reach equilibrium. The supernatants were collected via centrifugation and analyzed via a UV-visible spectrophotometer (Cary 100 model, Varian, Victoria, Australia) at a maximum wavelength ($\lambda_{max}$) of 610 nm. A calibration graph was necessary to evaluate the supernatant concentration.

The equilibrium removal capacity (mg/g) and the removal percentage (%) of the solids for BB-41 were detected using the formulas (Equations (S1) and (S2)).

### 4.4. Regeneration Procedure

The treatment of spent adsorbent with oxone and cobalt solution was adopted due to its simplicity, and it did not generate an additional pollutant in the environment. The procedure was reported somewhere else [25]. Briefly, fresh spent adsorbent was obtained after treatment with a solution of BB-41 ($C_i$ = 200 mg/L) for 4 h, and then separated via centrifugation, and added to a mixture of oxone and cobalt nitrate solution. The sample was stirred for 1 h, and then collected and added again to a fresh solution of BB-41. The same procedure was repeated 7 times as described above without changing the mixture of oxone and cobalt solution.

*4.5. Characterization Techniques*

To identify the crystalline phases in the as-prepared samples, the PXRD patterns were collected on a Bruker Advance D8 diffractometer (Karlsruhe, Germany) using Cu Kα. Thermogravimetric (TGA) and differential thermal (DTA) analysis were operated in the range of 25 to 900 °C on a TG-DTA, SDT 2960, TA instruments, (New Castle, DE, USA). The surface morphologies and elemental analysis were performed via scanning electron microscopy (SEM), a Jeol model, JSM-6700F (Tokyo, Japan), equipped with ESCA oxford instruments. The FTIR spectra were gathered on a Shimadzu spectrophotometer (Tokyo, Japan) using KBr pellets. The solid $^{29}$Si Magic angle spin ($^{29}$Si MAS NMR) spectra were collected on a Bruker 400 spectrometer (Karlsruhe, Germany). A 4 mm MAS probe head was used with sample rotation rate of 4.0 KHz. Micrometrics ASAP2040 (Ottawa, ON, Canada) was utilized to obtain information regarding the surface area, average pore size, and pore volume of the as-prepared samples. UV-VIS spectrophotometer from Varian (Cary 100 model, Varian, Victoria, Australia) was used to estimate the absorbance at maximum wavelength ($\lambda_{max}$ = 610 nm) in the supernatant.

## 5. Conclusions

Economic prosperity and development resulted in devastating implications on ecosystems via the contamination of natural effluents with industrial wastes. Based on the efforts of other studies, Na-magadiite with higher cation exchange capacities appear to be a plausible, economical, and practicable opportunity. Pure crystallized Na-magadiite was obtained from four different silica sources at a temperature of 150 °C for different periods of time from 1 day to 2 days. However, a mixture of Na-magadiite and Na-kenyaite was achieved at a longer reaction time. Pure Na-kenyaite was obtained at 170 °C for 2 days. The water and NaOH contents in the starting gel affected the types of the resulting phases, and minimum water and NaOH contents were required to obtain a pure Na-magadiite phase.

Interestingly, the morphology of the Na-magadiite depended on the silica source and other parameters, such as the size of the silica particles in the starting source. The rosette-like structure was detected using colloidal silica solutions at 150 °C for 2 days of hydrothermal treatment. This investigation has shown that identical synthesis runs resulted in similar crystalline products.

The removal properties of pure Na-magadiite materials were investigated using an artificially polluted water with BB-41 dye. The maximum removal capacity was influenced by the morphology of the materials, with a maximum of 220 mg/g for Na-magadiite with a rosette or cauliflower morphology. The pH after washing the solid samples after hydrothermal treatment played an important role in their ability to remove BB-41 dye molecules. A maximum removal percentage of 95% was achieved at pH values higher than 8. The regeneration studies indicated that these materials could be used after seven cycles, with a certain reduction in the removal percentages from 26 to 42%. The structure of Na-magadiite was not altered during this process. The design of a single-stage batch adsorber was proposed using the Langmuir model and mass balance equations. The required masses to reduce fixed volumes of contaminated water depended on the intended reduction percentages and the maximum removal capacities of the Na-magadiites used.

**Supplementary Materials:** The following supporting information can be downloaded at: https://www.mdpi.com/article/10.3390/inorganics11110423/s1, Figure S1: PXRD patterns of samples prepared from colloidal silica for different periods of times at 150 °C. * corresponds to crystalline quartz phase; Figure S2: PXRD patterns of samples prepared from Ludox-HS40 for different periods of times at 150 °C. * corresponds to crystalline silica phase; Figure S3: FTIR spectra of samples prepared from fumed silica at 150 °C for different periods of times; Figure S4: TGA (left) and DTG (right) features of products prepared from fumed silica at different periods of time and at 150 °C; Figure S5: PXRD patterns of MAG-FS calcined at different temperatures. The sharp reflection at 840 °C corresponds to quartz phase; Figure S6: Deconvolution of resonance peaks of Na-magadiites prepared from different silica sources. (a) MAG-HS, (b) MAG-AS, (c) MAG-CS, and (d) MAG-FS; Figure S7: $^{29}$Si MAS NMR spectra of materials prepared from colloidal silica at 150 °C for different

periods of time, d = day (s); Figure S8: SEM micrographs of materials prepared from Ludox-AS40% for different period of times at 150 °C.

**Author Contributions:** Conceptualization, A.M.A. and F.K.; data curation, S.R., S.A.P. and H.A.D.; formal analysis, F.K., H.A.D. and S.R.; funding acquisition, A.M.A. and F.K.; investigation, F.K., S.A.P. and R.A.-F.; methodology, F.K., S.R. and H.A.D. resources, F.K., S.R., H.A.D., A.M.A. and H.O.H.; supervision, F.K. and S.R.; validation, S.A.P., H.A.D., H.O.H. and R.A.-F.; writing—original draft, F.K., R.A.-F., H.A.D. and S.A.P. writing—review & editing, F.K., S.R., H.A.D. and H.O.H. All authors have read and agreed to the published version of the manuscript.

**Funding:** This research received no external funding.

**Data Availability Statement:** Data are available on request.

**Acknowledgments:** The researchers wish to extend their sincere gratitude to the Deanship of Scientific Research at the Islamic University of Madinah for the support provided to Post-Publishing Program 2. The authors would also like to thank Bernd Marler from the Institute of Geology, Mineralogy, and Geophysics, University of Bochum, Germany for providing the text file of the PXRD pattern of Na-kenyaite.

**Conflicts of Interest:** The authors declare no conflict of interest.

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
