# Peer review of "Parameters Synthesis of Na-Magadiite Materials for Water Treatment and Removal of Basic Blue-41: Properties and Single-Batch Design Adsorber"

_inorganics, doi:10.3390/inorganics11110423_

Round 1

Reviewer 1 Report

Inorganics Journal

Title: Parameters synthesis of Na-magadiite materials for water treatment: removal of Basic Blue-41: Properties and single batch design adsorber

Major revision

1-Abstract is poor written:

·       please clarify the utilized temperatures and periods of time

·       Various silica source was not selected in a systematic approach? I cannot understand

·       More details or results should be added to improve the abstract section

2- Characterization section should be moved to be below the synthesis section

3. The novelty in introduction section is not clear (i.e., Less literature has been reported. In the introduction section. You must compare the present study with the previous one and find the research gap.

4. Characterization results need more discussion

5. Figures in article must be improved

6. Modeling process is too bad and needs more discussion, particularly through the nonlinear method

7.  Implication of this work and a comparison between the proposed method and other methods should be provided in the manuscript.

8. Adsorption mechanisms are not reported

9. Based on the mentioned points, conclusion should be re-written 

10. English needs to be polished by a native English speaker.

must be improved

Author Response

The authors thank the reviewer for his comments, and they have tried all their bests to modify their manuscript accordingly,

Reply to Reviewer 1`:

1-Abstract is poor written:

  • please clarify the utilized temperatures and periods of time

This information was added in the abstract

  • Various silica source was not selected in a systematic approach? I cannot understand
  • Since this phrase has caused a confusing, so it was deleted

More details or results should be added to improve the abstract section

Extra results were added in the abstract.

2- Characterization section should be moved to be below the synthesis section

The authors have checked the order , and it was placed as mentioned

  1. 3. The novelty in introduction section is not clear (i.e., Less literature has been reported. In the introduction section. You must compare the present study with the previous one and find the research gap.

The synthesis of Magadiite in the laboratory was added, and the factors that affect the synthesis of pure magadiite is reported. Some references were added.

The novelty was added, and may be it was not clear because the authors have added extra information on the Basic Blue-41.

  1. Characterization results need more discussion

The authors have rechecked the results and they added more discussion when it was needed.

  1. Figures in article must be improved

The authors have tried to get good figures, and they were inserted directly from Sigmaplot software

  1. Modeling process is too bad and needs more discussion, particularly through the nonlinear method

The aim of the work was to estimate the maximum removal capacity of the prepared samples. In general the linear equation of Langmuir is the most  used one . In addition, the reported data in Table  5 was estimated from the linear form .

Nevertheless, additional information was added in the text.

The equation was not presented in the text, because this equation is very known,

  1. Implication of this work and a comparison between the proposed method and other methods should be provided in the manuscript.

The authors have tried to understand the meaning of this comment. The reviewer has mentioned the proposed method. It was not clear which method? Is it regeneration, or what? The authors apologize for that.

However, in the regeneration paragraph, the authors have added more details . The used method did not alter the structure of the Na-magadiite, and  the solution of oxone-coblat nitrate was used during the regeneration study, without changing it.

  1. Adsorption mechanisms are not reported

The authors agreed with the reviewer to describe the adsorption mechanism. The authors did not finish yet this study, and they were interested in the removal capacity.

This comment will be undertaken in the near future when the magadiite will be modified with heteroatoms.

In the literature, different mechanisms were proposed, and the mainly one was based on electrostatic interaction between the dye and the surface of the adsorbent.

  1. Based on the mentioned points, conclusion should be re-written 

The conclusion was rechecked and some results were added

  1. English needs to be polished by a native English speaker.

Before submission, the majority of the manuscript was edited by mpdi English service, and the authors have mentioned this comment to the letter for the editor, and the job reference was added too.

Reviewer 2 Report

In this work, the authors reported the synthesis and characterization of Na-magadiite materials, and investigated the effect of silica sources, sodium hydroxide, water contents on structure and morphology of the Na-magadiite materials. The influence of BB-41 initial concentration, Na-magadiite dosage, pH, silica sources, and morphology shape on adsorption properties was discussed. The optimal adsorption of the material was achieved. In my opinion, this manuscript can be considered for publication after major revision.

Some comments to the Authors:

1. For morphology characterization of the materials, EDS spectrum should be supplied.

2. For the effect of pH, the Zeta need to be investigated.

3. After 7 cycles, the structure stability of Na-magadiite materials should be characterized with XRD.

4. The adsorption mechanism need to be analyzed using XPS.

No

Author Response

The authors thank the reviewer for his comments

 Some comments to the Authors:

  1. For morphology characterization of the materials, EDSspectrum shouldbe supplied.

EDS spectrum for selected samples was added as figure 12.

  1. For the effect ofpH, the Zeta need to be investigated.

The authors have tried to find the devise to measure the Zeta potential, but It was difficult to find it in our area. The authors agrred with thereviewer and it will be interesting to do it. This study will be planned in th near future.

  1. After 7 cycles, the structure stability of Na-magadiite materials should be characterized with XRD.

The authors agreed with the reviewer, the XRD patterns were measured, and they were similar with slightly decrease in intensity of the major reflections. The data confirmed that the structure was maintained after these seven weeks, and thus the stability of the used materials during the regeneration process.

The authors have added this comment in the text.

  1. The adsorption mechanism need to be analyzed using XPS

The authors agreed with the reviewer to describe the adsorption mechanism. The authors did not finish yet this study, and their concern was to estimate the maximum removal capacity of these materials. 

This comment will be undertaken in the near future when the magadiite will be modified with heteroatoms.

In the literature, different mechanisms were proposed, and the mainly one was based on electrostatic interaction between the dye and the surface of the adsorbent.

This technique was not available in our  university and in the local universities.

Round 2

Reviewer 1 Report

The article was improved

Reviewer 2 Report

The authors have well revised this manuscript, I agree to recommend publication.

No